A proposed reconstruction method of a 3D animation scene based on a fuzzy long and short-term memory algorithm

Zhou Ming shanyuzm@163.com
Zhou Ming Anhui Sanlian College , Hefei , China
Shuja Junaid
Electronic publication date: 2024 Feb 29
Publication date: 2024
Volume: 10
Electronic Location ID: e1864
Received 2023 Nov 28; Accepted 2024 Jan 18
Copyright: ©2024 Zhou
Copyright year: 2024
Copyright holder: Zhou
License: This is an open access article distributed under the terms of the Creative Commons Attribution License, which permits unrestricted use, distribution, reproduction and adaptation in any medium and for any purpose provided that it is properly attributed. For attribution, the original author(s), title, publication source (PeerJ Computer Science) and either DOI or URL of the article must be cited.
License URL: https://creativecommons.org/licenses/by/4.0/

Keywords: Temporal neural network, 3D animation, Scene reconstruction, Deep learning

Funding: The authors received no funding for this work.

==============================
With the development of computer technology leading to a broad range of virtual technology implementations, the construction of virtual tasks has become highly demanded and has increased rapidly, especially in animation scenes. Constructing three-dimensional (3D) animation characters utilizing properties of actual characters could provide users with immersive experiences. However, a 3D face reconstruction (3DFR) utilizing a single image is a very demanding operation in computer graphics and vision. In addition, limited 3D face data sets reduce the performance improvement of the proposed approaches, causing a lack of robustness. When datasets are large, face recognition, transformation, and animation implementations are relatively practical. However, some reconstruction methods only consider the one-to-one processes without considering the correlations or differences in the input images, resulting in models lacking information related to face identity or being overly sensitive to face pose. A face model composed of a convolutional neural network (CNN) regresses 3D deformable model coefficients for 3DFR and alignment tasks. The manuscript proposes a reconstruction method for 3D animation scenes employing fuzzy LSMT-CNN (FLSMT-CNN). Multiple collected images are employed to reconstruct 3D animation characters. First, the serialized images are processed by the proposed method to extract the features of face parameters and then improve the conventional deformable face modeling (3DFDM). Afterward, the 3DFDM is utilized to reconstruct animation characters, and finally, high-precision reconstructions of 3D faces are achieved. The FLSMT-CNN has enhanced both the precision and strength of the reconstructed 3D animation characters, which provides more opportunities to be applied to other animation scenes.

Introduction

One way human beings access external information is through vision, which is screened, integrated, and analyzed through the brain system so that humans understand and feel the world around them. Research on computer vision, which enables machines to extract and analyze information from vision-based data quickly, has received great attention, similar to how humans achieve a reasonable understanding of external environment information.

A three-dimensional (3D) reconstruction is one of the substantial fields in computer vision, and its related technologies are widely implemented in film animation, robot navigation, medical imaging, relic protection, and many other fields (Wu et al., 2021). Scene simulation is an important research field of computer graphics, which is widely implemented in 3D video games, special effects, computer animation, military simulation, virtual reality, and augmented reality. 3D face reconstruction technologies and face animation are important research in computer graphics and focus on constructing 3D animation scenes (Casas et al., 2019). The conventional 3DFR technology mainly relies on costly 3D scanning devices, requires post-manual processes that take a long time, and necessitates scanned character subjects to maintain a fixed posture for a significant time. Moreover, hardware systems that enable running computations on a large scale have been developed recently, especially the development of distributed computing, which enables GPUs to generate efficient solutions.

In recent years, academic research and industry-based implementations have made great breakthroughs in detection and identification processes. The related research on the human body, especially the human face, has become a widely concerned research topic, and the scenes are also commonly employed in real life. The demand for 3D face models in face recognition, ARR technology, game animation, medical beauty, and other fields has increased. For example, 3DFR and face animation are two important research fields in computer graphics. Due to the complex spatial structure and rich fine attributions, 3D face reconstruction and face animation technologies have grown to become pivotal areas in the generation of computer graphics.

A face is the most recognizable part of an individual. Still, the characteristics of human faces in 2D space are generally constrained by uncertain factors such as lighting, facial expression, angle of face, occlusion, etc. Therefore, studying human faces in 3D space has become a breakthrough in the next step of development (Wang & Zhai, 2019).

Keeping in mind that face videos help reconstruct 3D faces in 3D face animation scenes by making full use of potential face geometric structure and apparent information, and considering the correlation of different images in time, thus mastering the properties of face features, finally recovering the real 3D face geometric model as close as possible is an open problem. The 3D face model generated by employing the proposed network (FLSTM-CNN) could reflect certain personalized features and result in certain effectiveness and robustness. Moreover, it can bring significant convenience to music users, so it presents a substantial research area and wide implementation possibilities.

The manuscript is sectioned as follows: ‘Introduction’ introduces the research background and motivation, the predominant methods, and the effects of the reconstruction. ‘Preliminary’ introduced the sequential deep neural network and the 3D reconstruction algorithm. ‘The proposed method’ introduced the proposed model, modeled the human face, and evaluated the reconstruction effect. ‘Experimental results and discussion’ mainly summarizes the conducted research and presents the future direction.

Preliminary

A 3DFR has been one of the popular study areas in computer vision. Many enterprises, institutes, and universities have invested a lot of funds and resources and generated a lot of distinct algorithms and methods to improve the impact of a 3DFR. The robustness of the algorithms has advanced greatly (Mai et al., 2018). Moreover, continuous research progress and technological innovations have enhanced several methods of 3DFR that could be split into two classes: 3DFR employing conventional approaches and deep learning algorithms (Jin et al., 2017).

Parke (1982) was the first to propose a face parameterization model that employed the grid representation of a face utilizing about 400 vertices and 250 polygons by changing the structure and action of the grid surface, implementing rotation, translation, and scale, which makes face shapes change. Thus, the characteristics of personalized faces were initially realized between different individuals (Becker & Fregosi, 2017). Muscle models have emerged to cope with the problem that the proposed parametric models cannot accurately represent facial features and expressions. Zhao (2001) connected the elastic features of the facial skin vertices, utilized force on the facial muscles to realize the representation of the facial features, and established a 3D facial model (Hernandez et al., 2017). A muscle model that can be employed to simulate the process of facial expression changes was also proposed by Waters (1987). The model caused the deformation of muscles by utilizing the force when the expression changed and made the face grid represented by the polygon deformation. This model could align the expression information with the real facial expression (Merras et al., 2017). Lee et al. (2022) proposed to employ a high-precision 3D face scanner to attain 3D point cloud data and facial texture information with depth information. Afterward, the corresponding 3D face algorithm is constructed according to the obtained information. The reconstruction results obtained by this method were relatively accurate, but the equipment was expensive, and the data acquisition was limited (Shirowzhan et al., 2020). Blanz & Vetter (1999) first proposed the famous 3D deformation model that led to the implementation of statistical analysis, which is of landmark significance for 3DFR (Turan, 2017).

Methods employing deep learning algorithms (DLAs) have advanced swiftly and provided superior performance in both computer vision and image processing. Many scholars have also started to employ DLAs to examine 3DFR. Zhu et al. (2019) proposed a 3D face alignment network that converted the dense 3D face algorithm into a 2D face image through CNNs to improve face alignment performance at large poses. To resolve the issue of the unnoticed significant points of the face under the big posture (Tu et al., 2020), a robust approach for regression-separable 3D deformation models (3DFDM) was designed by Tran et al. (2017). The researchers suggested employing a CNN to directly regress the shape and texture parameters from the input photos and then fit the 3D face approach. The accurate 3D face approach could be reconstructed by inputting a single image and was faster when 3DFDM parameters were estimated. Tran & Liu (2018) proposed a nonlinear 3DFDM method to model shapes or textures through deep neural networks (DNNs) (Guo et al., 2018). The conventional 3DFDM employed linear basis approaches for shapes or textures, which were trained on 3D face scans and associated 2D images. However, this method did not require 3D scanning by implementing weak supervision. It could be trained from naturally unconstrained 2D face images that align with nonlinear changes such as facial expressions and gestures. So, the original image could be better reconstructed with a stronger representation, and the performance of related tasks was further improved (Feng et al., 2018b). The voxel regression network (VRN) proposed by Jackson et al. enabled the 3DFR of 2D face images utilizing arbitrary facial poses and expressions without constructing or fitting 3DFDM parameters during training and testing stages (Yueyang, Wan & Junkaiyi, 2019), respectively. This method could obtain an end-to-end representation of 3D facial geometry, while it had the problem of a large overall computational burden and limited resolution in reconstructions. Feng et al. suggested an end-to-end position mapping regression network (PRN) that could realize 3DFR and dense face alignment in two tasks by designing a UV position mapping as a 2D representation method, which was implemented to record the UV space. Thus, it enabled 2D image storage for 3D face cloud coordinates. Each UV polygon contained the corresponding semantic information (Zhang et al., 2021). Richardson, Sela & Kimmel (2016) reconstructed the 3D face geometry by CNNs regression to the 200-dimensional identity vector and 84-dimensional expression vector. They then aligned the recovered face mask to the input image and optimized the face shape by utilizing the network to obtain the reconstruction results with rich, detailed features. With the utilization of an unsupervised training method combined with the 3DFDM proposed by Genova et al. (2018) the face recognition network was innovatively introduced into the training loss function based on the editor and decoder models so that the generated 3D faces could retain the individual face features of the input pictures (Trâń et al., 2018).

The literature suggests that the demand for implementations constantly increases, and employing 2D images to recover 3D face data is a substantial qualitative problem (Peng & Tian, 2017). Therefore, the current research work still faces many challenges. This is mainly due to the serious problems of available single-view 3DFR methods when applied in real scenes. The 3D face shape estimation implementing a single image is either unstable or could be different when different photos of the same object are presented, or the estimated 3D face shape is overgeneralized but not applicable. However, collecting large amounts of 3D scanned facial data simultaneously is laborious and impractical (Rui et al., 2018). Instead, the large quantity of 2D face data could help better. Therefore, to improve the learning capability and strength of the algorithms, it is important to examine the 2D face reconstruction model. First, a 3D scene reconstruction algorithm based on temporal deep learning, whose basic framework is shown in Fig. 1, is presented. With the utilization of the temporal deep neural network, the video-sampled image sequence is learned, and then the face is shaped by the CNN to reconstruct the 3D face animation.

Figure 1 A reconstruction method of a 3D animation scene based on time series deep learning.

The Proposed Method

A brief introduction to RNN and LSTM algorithms

When using a video to extract face images, a series of time series images are obtained. In deep learning algorithms, neural networks that can crunch time-dependent data are the best way to process time series data. A recurrent neural network (RNN), a fully connected neural network with self-loop feedback processing time series data, is presented in Fig. 2. The result of its neuron is passed onto itself at the next time point, and it also outputs a hidden layer state for the current layer to be implemented when the sampled observations are processed at the next time point.

Figure 2 The structure of an RNN.

The input, hidden, and output layer vectors are denoted by x, s, and o, respectively. U, V, and W denote the coefficient matrices of the input layer and hidden layer, the hidden layer to the output layer, and weights, respectively. The score s of the hidden layer is obtained from the current moment input x and the hidden layer s of the previous moment. As shown in Fig. 1, the st at time t is decided by xt and st−1.

The forward propagation of the RNN is employed to predict the next value. For example, yt represents the true value at time t, and xt denotes the input for time t, then yt ^ represents the predicted value at time t. st is the hides of the state at time t. The output is denoted by ot.

(1) st=∅Wxxt+Wxxt−1+bs

(2) ot=W0xt+b0

(3) yt ^=σot

where ∅ represents the activation function that is tanh, WX denotes the coefficient matrix of the input layer to the hidden layer; WS denotes the coefficient matrix between the hidden layers; WO is the weight matrix from the hidden layer to the output layer; bS is the offset item; bO is the bias item; σ is an active function, generally called the softmax function. For example, at time t3, the loss function is defined by

(4) L3=12y3−o32

The RNN’s backpropagation process utilizes the stochastic gradient descent method to guide the bias term in the network and continuously update it to minimize the loss value L. However, when the RNN gets trained, gradient disappearance or explosion (GDE) could occur. So, the long and short-term memory network (LSTM) is implemented to overcome this issue.

The LSTM, initially suggested by Hochreieter & Schmidhuber (1997), is based on the improvement of the RNN and can recall both long and short-term information. The LSTM is composed of an LSTM unit with a chain structure and adds thresholds to the algorithm’s input, forgetting, and output layers, respectively so that the coefficient of its cycle alters, permitting the network to forget the currently gathered information, thereby preventing GDE. Forget, input, and output gates denoted by f, i, and o, respectively, in the LSTM unit are input from the hidden state ht-1 of the prior moment and the current time step input xt, and their activation mappings are depicted in Fig. 3.

Figure 3 Activation functions used in the RNN-LSTM.

The threshold mechanism of the LSTM network allows it to stash information for a long time, thus preventing GDE. Input, forgetting, and output gates are placed in each unit, and all employ the sigmoid activation function to control the transmission of information in the network, so they assign certain information at the current moment and then to the information required by the network at the following time. The value domain of all the gates is {0,1}, where 0 denotes all representative information forgotten, and 1 is the opposite. Input, forget, and output gates control which information remains at the current time step, which prior time step information must be forgotten, and which current time step information must be output, respectively.

(5) Ct=tanhWxcxt+Whcht−1+bc

(6) Ct=Ct−1∗ft+δt∗it

(7) ht=tanhCt∗ot

The predicted score of the model at time t is delineated by

(8) yt ^=σWht+C.

when compared with the RNN’s solution process to derive bias parameters, multiple branches appear in the LSTM, accumulating in every successive step, and the calculation parameters are directly or indirectly related to the matrix Wxf. As it increases, the cumulative term increases along with the gradient calculation. Due to these additive terms, the vanishing gradient problem in the RNN is effectively resolved by LSTM.

3D face deformation model (3DFDM)

Network regression parameters are usually employed as the backbone of a time-dependent neural network. The coefficients of a 3D face deformation model (3DFDM) complete 3DFR by utilizing a CNN to input extracted images’ face features and then map the feature vector to 3DFR parameters to finalize the whole reconstruction process (Lium et al., 2021). A 3DFDM-based 3DFR method is implemented to determine the shape robustness to any face posture, namely, the different face postures of the same identity employing the network regression parameters are based on similar shape parameters. In other words, the 3D face form of the same object recovering different attitudes of the 3D face shapes should be close to its original 3D face shape. If a deviation occurs, the algorithm constraints are employed to make it as close as possible to the 3D face shape. Therefore, the problem is converted into a clustering problem. To address the problem, two different objects were selected from the 300W-LP dataset, reducing the problem to dichotomous problems, as depicted in Fig. 4.

Figure 4 The schematic diagram of the 3DFDM parameter attribute decomposition network.

The 3D face shapes recovered from the 3DFDM parameters predicted by the two objects may differ. With the inclusion of the scale, shape, or expression aspects, the 3DFDM parameters are projected into a 2D space. Then, the 3DFDM parameter distribution of different poses would become more discrete. However, the 3D face shapes reconstructed by 3DFDM parameters are more consistent after shape constraints are embedded. The projection of the 3DFDM parameters of different poses in the 2D plane is also relatively concentrated with increasing interclass distance. The robustness of 3D face shapes is effectively resolved with the research results of 2D face images; that is, the predicted 3DFDM parameter vectors are clustered while reducing the intra-class length and increasing the inter-class difference.

When constructing a 3DFDM, the modeling is based on the available database of faces, and all the 3D face models in the database are represented by a triangular grid composed of the same number of vertices and triangular face slices so that the same index point has the same semantic information. The 3D facial geometry of a human face can be expressed in the vector form in Eq. (9). (9) S=x1,y1,z1,Lxn,yn,znT∈R3n

where S and n represent the shape vector, and the total vertex numbers that constitute the face model, respectively. There are m numbers of different 3D face samples in the known face database. Each of these can be represented by the shape vector S. Therefore, a 3D face geometry of a new face can be obtained through a linear combination of the shape vectors of the m different personal face samples in the knowledge database presented in Eq. (10). (10) Snewmodel= ∑i=1mαiSi.

With the alteration of the shape coefficient of a human face’s mouth structure, a new geometric model of a human face can be generated.

Deep learning algorithm based on fuzzy temporal neural network

The process of face reconstruction implements CNN to master features. Feature extraction is performed by employing a fuzzy LSTM (FLSTM) network proposed by Pattanayak, Sangameswar Vodnala & Das (2022) as the backbone network of the FLSTM-CNN while replacing the last two layers of the network with two parallel fully connected layers (FCL). The FLSTM’s algorithm is given as follows:

Algorithm: The suggested FLSTM

Input the time series observations as y = [y1, y2, ⋯ ⋯, yn]

T, length of training set (ℓtr), length of validation set (ℓ), and length of test set (ℓt).

Output Forecasted score.

1.  Calculate b0 and b1 for the time series observations b1 = ∑ (ti − ∑ ti n i =1 n) ∗ n i =1 (y i − ∑ y i n i =1 n) ∑ (t i − ∑ t i n i =1 n) 2 n i =1 b0 = 1 n (∑ y i − b1∑ t i n i =1 n i =1)

2.  Calculate the trend score of each observation in the time series data. for t = 1 to n(size of TS) ut = b0 + b1 t end of for

3.  Compute the RTV scores i.e., V of each observation in the time series data for t = 1 to n (Size of TS) Vt = (y t u t ⁄) ∗ 100 end of for

4.  By using the RTV data’s median, it is split into P and Q subgroups.

5.  Calculate the absolute first difference Pd and Qd in both P and Q subgroups.

6.  Attain the mean of the Pd and Qd as Pv and Qv for both the subsets P and Q, respectively.

7.  Attain the deciding factor ϑ, ϑ = 10 floor(e) (Pv Q)

8.  Attain the NOIs i.e., k as follows k = ((min(V) − ϑ) − (max(V) + ϑ) ϑ) + ℓt × n

9.  Find the UOD as: U = (min(y) − δ, max(y)+ δ) where δ denotes the standard deviation.

10.  Attain the NOIs k of the time series data.

11.  Split the UOD U = {u e1, u e2, u e3, ⋯, u e w,} into k equal interval lengths and Calculate the midpoint mi (i = 1,2, ⋯, w) of each interval. // The time series data is fuzzified.

12.  for r = 1 to n (length of time series) if y r belongs to the partition uei fr (t) = i end of if end of for //Construct the FLR by employing LSTM

13.  FLSTM’s order is attained by utilizing the autocorrelation function.

14.  Employ Min-Max normalization to attain the normalized in sample time series observations f′ (t).

15.  Convert the f (t), into n − r patterns using sliding window protocol.

16.  Split the time series observations into training, validation, and test sets.

17.  Attain normalized forecasting outcome f′(t) utilizing the LSTM’s optimized parameters.

18.  Denormalize the forecasting outcome f(t)

19.  Attain the forecasted score using defuzzification of f(t) by employing a middle point fuzzy interval and attain the real forecasted score y ^.

The CNN implemented for feature learning is shown in Fig. 5. The FLSTM network is employed as the backbone of the FLSTM-CNN to extract features, while the last two layers of the network are replaced with two parallel fully connected layers (FCLs). One FCL represents a 62-dimensional vector of the 3DFDM parameters; the other fully connected layer outputs a 512-dimensional face feature vector for face recognition. As a sub-network of the FLSTM-CNN, the network structure has the same network structure as its symmetric CNN, sharing the network parameters mastered by the model.

Figure 5 The production planning process of the proposed algorithm.

In the manuscript, we assume the time series data to be fuzzy or fuzzified, so the fuzzy time series method is applied to DLAs. The generation operation of the proposed FLSTM-CNN algorithm is depicted in Fig. 5. First, the 3D capture software is employed to attain the 3D model observations of the face and the observations are fuzzified. Then, the data processing software is employed to extract attribution points of the 3D animations by implementing the FLSTM-CNN. The modeling software realizes the modeling realization of a 3D animation. The entire model is trained by minimizing the three loss functions.

The first loss function is employed to measure the precision by predicting the 3DFDM parameters, called the weighted prediction of loss function coefficients. The second loss function is employed to maintain the consistency of the face shape. When the image is input, the reconstruction parameters of similar face shapes are output. The third loss function is called the identity loss function, which ensures that the facial images with the same identity have a similar distribution in the feature space. Therefore, the loss function of the overall training is defined by. (11) t=t3d+tshp+tid.

Overall, we observed that currently popular 3D face deformation methods, namely, 3DFDM and DeFA, considered only the one-to-one reconstruction process but not the correlation or differential information of the input images in the training set, which led to the reconstruction results being unreliable for face identity representations.

The proposed fuzzy temporal CNN (FLSTM-CNN) implements a many-to-one training strategy to restrict the face geometry of the same person in different poses to the same shape while preserving the identity information of the recovered shape. The proposed algorithm could learn more robust 3DFDM parameters and obtain more discriminative features for identity representations.

The proposed algorithm employs the classical 3DFDM to recapture the 3D face shape from a single image, the same as the 3DFDM. Modifications are made on a theoretical basis to satisfy the requirements of the proposed algorithm. Based on the regression task algorithm utilizing the mean square error criterion to evaluate the accuracy of the predicted values, the failure of weighing the loss is considered a deficiency since network optimization is slow, and the effect of the model prediction is not good. The improved WPDC loss function is chosen as the evaluation function of the 3DFR accuracy combined with the contrast loss function, finalizing the 3D face model training and optimization processes. The heavy-distance cost function is trained for the accuracy of the reconstruction of the 3D face model according to the importance of each predicted 3DFDM parameter.

Experimental Results and Discussion

Experimental dataset and preprocessing

In the construction process of the 3D face model database employing the FLSTM-CNN, the video data of a user is employed to gather different 2D face images with distinct facial gestures and facial expressions. Then the 3D face model corresponding to the image is obtained as training data. A scanning device of a 3D model (hardware equipment: Structure Sensor) is employed to scan the 3D shape of the user’s face and edit the 3D face model grid (3D-Mesh). The face template database for the user’s face features is implemented to run further smooth processing. The 3D scanning device can easily and quickly obtain the 3D face model in real-world applications. Still, the scanned 3D model contains large noise, and the topology cannot be effectively unified, which causes some difficulties in the automation process (Feng et al., 2018a).

The process of feature acquisitions first implements the template model and the rigid registration of the target model, namely, rotation, translation, and zoom, which leads to overlapping the centers of the two models. In the 3D face registration model, a model often has certain semantic feature information regarding faces. For the 3D face template model and the same semantic information, a simple semantic correspondence is established between the template and target models based on the corresponding rigid alignment template model of the rotation transformation matrix and target model, respectively. Multiple iterations are run to optimize it. The matching registration of the two 3D models means finding the mapping of the 3D template model and the 3D target model, respectively. Therefore, the mapping must fully express the corresponding vertices with the same semantic information as the two models. The matching registration outcomes show that the 3D template model has the individual characteristics of the 3D target model and the corresponding facial expression features, respectively (Thakrar et al., 2017). Considering the type of target model and registered scene, different matching registration algorithms often show a completely different effect. To build a scene model, the matching registration process is relatively simple. In contrast, the 3D face model has richer facial detailed features and scans the 3D model whose noise is larger, so it needs multi-step continuous iterative optimization.

Although the above 3D face model database and the corresponding data construction method contain different individuals and expressions, the database expresses detailed facial expressions that could be utilized in computer graphics, vision, and many other fields. The database, composed of 3D facial expressions of 150 individuals of different ages, races, and genders, utilized in the manuscript is called Face-Warehouse (Liu et al., 2018). Each has a variety of rich expressions, such as mouth, smile, anger, frown, pout, mouth, and so on. Since the employed 3D face model has the same topology, a 3D tensor containing the database in the experimentation is implemented to construct a bi-linear 3D face model with both individual and expression properties.

The 3D face model data plays a key role in developing practical applications. With the utilization of the available 3D model data, 3D reconstruction and expression animation can be developed. An accurate 3D model generally needs to go through four stages to extract face feature points, generate a face model, adjust the face model, and display the model. However, the available 3D model datasets often have certain limitations in the breadth of data. Systematic production based on available data expands the character library in the breadth of the dataset and enables fast and precise production of individual virtual characters. At the same time, it enriches the facial gesture expressions of the character database in the depth of the data set, providing rich data for computer graphics and visions. Hence, virtual reality, interactive education, and face-driven animation technology utilizing virtual characters provide directly feasible technology solutions for the industry.

Time-dependent deep learning-based scene reconstruction algorithm

Several different algorithms mentioned previously are implemented to reconstruct face scenes, and the effect is shown in Fig. 6. Although the face shape reconstructed by the PRN method is inconsistent with the mouth shape, the suggested algorithm results in the constructed mouth shape with no change. The comparison outcome suggests that the suggested algorithm is more robust than PRN reconstruction when reconstructing 3D face shapes based on the different poses. The implemented deep learning framework is called Pytorch, employing the SGD as the optimizer and implementing the exponential decay learning rate strategy expressed by Eq. (12). (12) lr=λx∗0.001epoach+1−sdte±sd.

Figure 6 Data renderings.

Due to the limitations of the test data, the K-fold cross-validation algorithm is employed to test the accuracy of the face recognition algorithm. However, when the dataset is small, this method can be employed to test the accuracy of the face recognition algorithm to improve data utilization effectively. K-fold cross-validation first splits the observations into K subsets, as shown in Fig. 7. For example, when K = 10, the observations are divided into ten equal subclasses, and nine subsets are employed for the training set; the last one is kept as the test set. Then, the average of all recognition accuracy rates based on the recognition accuracy rate of each subset is utilized to compute the final recognition rate.

Figure 7 The outcomes of k-fold cross-validation.

The manuscript verifies the performance of the suggested attribute decomposition learning network for the 3D face alignments on two datasets, called AFLW and AFL, and W2000.3D, respectively. The test results and the quantitative comparison with other conventional approaches are shown by utilizing the cumulative error distribution curve (CED) presented in Figs. 8 and 9, respectively.

Figure 8 The CED curves of the 3D face alignment outcomes on the AFLW dataset.

Figure 9 The CED curves of the 3D face alignment outcomes on the AFLw2000 dataset.

The experimental results show that the 3D face scene reconstruction method based on FLSTM-CNN could generate better outcomes.

Conclusion

A 3DFR employing single pictures makes it difficult to recognize faces when computer graphics and vision technologies are implemented. Face-positive and face-animation applications are of great significance and suffer from this issue. Recent studies have mainly focused on the CNN-based face regression models implementing the coefficient of 3DFDM to reconstruct 3D face and alignment tasks. However, reconstruction methods only consider a one-to-one reconstruction process, regardless of the correlation or differences in the input image. This leads to the model’s lack of sensitivity to identify face information or deal with excessive sensitivity to face posture. In addition, the lack of a 3D face data set limits the performance of the mentioned methods that lack the robustness of natural face images.

Based on the problems encountered in the literature, the fuzzy temporal neural network (FLSTM) is proposed to process the sequence of images and improve the original 3DFR algorithm (3DFDM). The video image frame algorithm is used to realize the 3DFR algorithm. Experiments designate that the suggested method could lower the number of images needed even though it has a higher accuracy and presents more robustness. Moreover, it can not only be applied to the reconstruction of the 3-D face animation but also to the reconstruction of other 3D animation scenes, which has strong practical value.

Time-dependent data could be treated with other fuzzy time series methods available in the literature to search for better outcomes. Future research will concentrate on the implementation and comparison of those methods.

Supplemental Information

Supplemental Information 1 Code

Additional Information and Declarations

Competing Interests

Author Contributions

Data Availability

The authors declare there are no competing interests.

Ming Zhou conceived and designed the experiments, performed the experiments, analyzed the data, performed the computation work, prepared figures and/or tables, authored or reviewed drafts of the article, and approved the final draft.

The following information was supplied regarding data availability:

The code is available in the Supplementary File.

The PK image is copied and edited from https://en.photo-ac.com/photo/24455124/a-boy-appealing-for-environmental-problems

One of the images used is available from https://en.photo-ac.com/photo/24455124/a-boy-appealing-for-environmental-problems

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
