# Peer review of "A proposed reconstruction method of a 3D animation scene based on a fuzzy long and short-term memory algorithm"

_PeerJ Computer Science, doi:10.7717/peerj-cs.1864_

## Round 0.1 · original submission · Major Revisions

The authors should incorporate all reviewers comments while submitting the revised article.

**Language Note:** The review process has identified that the English language must be improved. PeerJ can provide language editing services - please contact us at copyediting@peerj.com for pricing (be sure to provide your manuscript number and title). Alternatively, you should make your own arrangements to improve the language quality and provide details in your response letter. – PeerJ Staff

Reviewer 1 ·

Basic reporting

The article contributes to the available literature by fuzzing the LSTM model. However, some technical issues are detected and need to be fixed. We recommend a major revision.
The issues detected are presented as follows:
-All sections, including the abstract, should be checked regarding language and logical order.
-Why did the author need to adopt a fuzzified deep learning model (FLSTM-CNN)? What makes them use fuzzified data? Please provide more remarks and discussion.

Experimental design

-Did the authors compare the fuzzified version and the non-fuzzified version of the model to derive more insights? Please discuss it.
-Did the author run validations for the constructed model? Please discuss it.
-Which method did the author use in fuzzification? Please provide more discussion
-Which software is used, and what are the percentages of test and training data sets?
-Could the constructed method be applied to other modeling efforts directly? Please discuss more

Validity of the findings

-More discussion should be provided with the data sets, AFLW AFL, and W2000.
-Did the authors run any preprocessing and normalization steps?
-Which assessment metrics are used to evaluate the proposed model? Please discuss it.
-What do figures 8 and 9 tell us about? Please provide more discussion and remarks.

Reviewer 2 ·

Basic reporting

he itemized issues should be resolved properly. A major revision is required to improve the article.
 1.⁠ ⁠The abstract is very long and should be shortened.
 2.⁠ ⁠the abstract should summarize the conducted research by mentioning the research motivation, the data, the proposed method, and the contribution with a key finding. The current abstract should be modified based on those criteria.
 3.⁠ ⁠the number of citations is 3 in the introduction section. This is not enough. The number of citations and discussions should be increased.
 4.⁠ ⁠The research motivation and contribution of the research should be expressed in a paragraph in the introduction.
 5.⁠ ⁠This is extracted from the text: “3D deformation models (3DFDM) was designed by Tran et al.” One of type of citation should be adopted and used. All citations in the text should be checked and fixed.
 6.⁠ ⁠Some paragraphs are very long. They need to be shortened.
 7.⁠ ⁠All mathematical expressions should be cited and numbered in the text. Terminology should be explained just below the equations.

Experimental design

8.⁠ ⁠Which software is used to run the model?
9.⁠ ⁠How was the data fuzzified? Which type of fuzzy function is used in fuzzfying stage? Please discuss it.
10.What are the percentages of training and test data sets? Please discuss it.
11.What advantages are expected when fuzzy theory is applied to LSTM model? Please discuss it.

Validity of the findings

12.The conclusion section should be improved.
13.All abbreviation should be checked and fixed if not properly used.

Additional comments

NA

---

## Round 0.2 · accepted · Accept

The authors have made significant changes to address the reviewers' comments. The article is acceptable in its current form.

Reviewer 1 ·

Basic reporting

The changes are made efficiently

Experimental design

The experimental design seems more understandable now

Validity of the findings

As a whole, the paper looks good to be accepted.

Reviewer 2 ·

Basic reporting

All the concerns have been addressed. The paper can be accepted in its current state.

Experimental design

NA

Validity of the findings

NA

Additional comments

NA